# DeepRank: a deep learning framework for data mining 3D protein-protein interfaces

Nicolas Renaud[1], Cunliang Geng [1,2], Sonja Georgievska [1], Francesco Ambrosetti [2], Lars Ridder [1], Dario F. Marzella [3], Manon F. Réau[2], Alexandre M. J. J. Bonvin [2✉] & Li C. Xue [2,3✉]

Three-dimensional (3D) structures of protein complexes provide fundamental information to decipher biological processes at the molecular scale. The vast amount of experimentally and computationally resolved protein-protein interfaces (PPIs) offers the possibility of training deep learning models to aid the predictions of their biological relevance. We present here DeepRank, a general, configurable deep learning framework for data mining PPIs using 3D convolutional neural networks (CNNs). DeepRank maps features of PPIs onto 3D grids and trains a user-specified CNN on these 3D grids. DeepRank allows for efficient training of 3D CNNs with data sets containing millions of PPIs and supports both classification and regression. We demonstrate the performance of DeepRank on two distinct challenges: The classification of biological versus crystallographic PPIs, and the ranking of docking models. For both problems DeepRank is competitive with, or outperforms, state-of-the-art methods, demonstrating the versatility of the framework for research in structural biology.

[1] Netherlands eScience Center, Science Park 140, 1098 XG Amsterdam, The Netherlands. [2] Bijvoet Centre for Biomolecular Research, Faculty of Science - Chemistry, Utrecht University, Padualaan 8, 3584 Utrecht, CH, The Netherlands. [3] Center for Molecular and Biomolecular Informatics, Radboudumc, Greet Grooteplein 26–28, 6525 Nijmegen, GA, The Netherlands. ✉email: a.m.j.j.bonvin@uu.nl; me.lixue@gmail.com

Highly-regulated protein-protein interaction networks orchestrate most cellular processes, ranging from DNA replications to viral invasion and immune defense. Proteins interact with each other and other biomolecules in specific ways. Gaining knowledge on how those biomolecules interact in 3D space is key for understanding their functions and exploiting or engineering these molecules for a wide variety of purposes such as drug design[1], immunotherapy[2], or designing novel proteins[3].

In the past decades, a variety of experimental methods (e.g., X-ray crystallography, nuclear magnetic resonance, cryogenic electron microscopy) have determined and accumulated a large number of atomic-resolution 3D structures of protein-protein complexes (>7000 non-redundant structures in the PDBe databank (https://www.ebi.ac.uk/pdbe/) as of Sep. 2nd 2020). Numerous machine learning methods[4], and recently several deep learning techniques[5–7], have been developed to learn complicated interaction patterns from these experimental 3D structures. Unlike other machine learning techniques, deep neural networks hold the promise of learning from millions of data without reaching a performance plateau quickly, which is computationally tractable by harvesting hardware accelerators (such as GPUs, TPUs) and parallel file system technologies. Wang et al[5]. have trained 3D deep convolutional networks (CNNs) on 3D grids representing protein-protein interfaces to evaluate the quality of docking models (DOVE). Gaiza et al[6]. have recently applied Geodesic CNNs to extract protein interaction fingerprints by applying 2D ConvNets on spread-out protein surface patches (MaSIF). Graph Neural Networks (GNNs)[8], representing protein interfaces as graphs, have also been applied to predict protein interfaces[7]. Finally, rotation-equivariant neural networks have recently been used by Eisman et al. on point-based representation of the protein atomic structure to classify PPIs[9]. One outstanding illustration of the potential of deep neural networks in structural biology is the recent breakthrough in single-chain protein structure predictions by AlphaFold2[10–12] in the latest CASP14 (Critical Assessment of protein Structure Prediction round 14). Predicting the 3D structure of protein complexes remains however an open challenge: in CASP14 no single assembly was correctly predicted unless a known template was available. This calls for open-source frameworks that can be easily modified and extended by the community for data mining protein complexes and can expedite knowledge discovery on related scientific questions.

Data mining 3D protein complexes presents several unique challenges. First, protein interfaces are governed by physico-chemical rules. Different types of protein complexes (e.g., enzyme-substrate, antibody-antigen) may have different dominant interaction signatures. For example, some complexes may be driven by hydrophobicity, and others by electrostatic forces. Second, protein interactions can be characterized at different levels: Atom-atom level, residue-residue level, and secondary structure level. Third, protein interfaces are highly diverse in terms of shapes, sizes, and surface curvatures. Finally, efficient processing and featurization of a large number of atomic coordinates files of proteins is daunting in terms of computational cost and file storage requirements. There is therefore an emerging need for generic and extensible deep learning frameworks that scientists can easily re-use for their particular problems, while removing tedious phases of data preprocessing. Such generic frameworks have already been developed in various scientific fields ranging from computational chemistry (DeepChem[13]) to condensed matter physics (NetKet[14]) and have significantly contributed to the rapid adoption of machine learning techniques in these fields. They have stimulated collaborative efforts, generated new insights, and are continuously improved and maintained by their respective user communities.

Here we introduce DeepRank, a generic deep learning platform for data mining protein-protein interfaces (PPIs) based on 3D CNNs. DeepRank maps atomic and residue-level features calculated from 3D atomic coordinates of biomolecular complexes in Protein Data Bank[15] (PDB, www.wwpdb.org) format onto 3D grids. DeepRank applies 3D CNN on these grids to learn problem-specific interaction patterns for user-defined tasks. The architecture of DeepRank is highly modularized and optimized for high computational efficiency on very large datasets up to millions of PDB files. It allows users to define their own 3D CNN models, features, target values (e.g., class labels), and data augmentation strategy. The platform can be used both for classification, e.g., predicting an input PPI as biological or a crystal artifact, and regression, e.g., predicting binding affinities.

In the following, we first describe the structure of our DeepRank framework. To demonstrate its applicability and potential for structural biology, we apply it to two different research challenges. We first present the performance of DeepRank for the classification of biological vs. crystallographic PPIs. With an accuracy of 86%, DeepRank outperforms state-of-the-art methods, such as PRODIGY-crystal[16,17] and PISA[18], which respectively reach an accuracy of 74 and 79%. We then present the performance of DeepRank for the scoring of models of protein-protein complexes generated by computational docking. We show here that DeepRank is competitive and sometimes outperforms three state-of-the-art scoring functions: HADDOCK[19], iScore[20,21], and DOVE[5].

## Results

**Description of DeepRank.** DeepRank is built as a Python 3 package that allows end-to-end training on datasets of 3D protein-protein complexes. The overall architecture of the package can be found in Supplementary Note 1 together with details regarding its implementation. The framework consists of two main parts, one focusing on data pre-processing and featurization and the other on the training, evaluation, and testing of the neural network. The featurization exploits MPI parallelization together with GPU offloading to ensure efficient computation over very large data sets.

Data pre-processing and featurization:

1. Feature calculations. Starting from the PDB files that describe the 3D structures of protein-protein complexes, DeepRank leverages pdb2sql[22], our fast and versatile PDB file parser using Structured Query Language (SQL), to identify interface residues between the two chains. Interface residues are by default defined as those with any atoms within a 5.5 Å distance cutoff (configurable) from any atom of another chain (Fig. 1A). The atomic and residue-based features presented in Table 1 are by default calculated, but users can easily define new features and include them in their feature calculation workflow.

2. 3D grid feature mapping. DeepRank maps the atomic and residue features of the interface of a complex onto a 3D grid using a Gaussian mapping (see Methods). The grid size and resolution can be adjusted by users to suit their needs. Fig. 1A illustrates the mapping process for a residue-based feature (see the Methods section for explanations). Thanks to this gaussian mapping, each feature has a non-local effect on the 3D feature grid, contributing to a multitude of grid points. This feature mapping of the PPIs results in a 3D image where each grid point contains multiple channel values corresponding to different properties of the interface. Several data augmentation and PPIs structure alignment strategies are available to enrich the dataset.

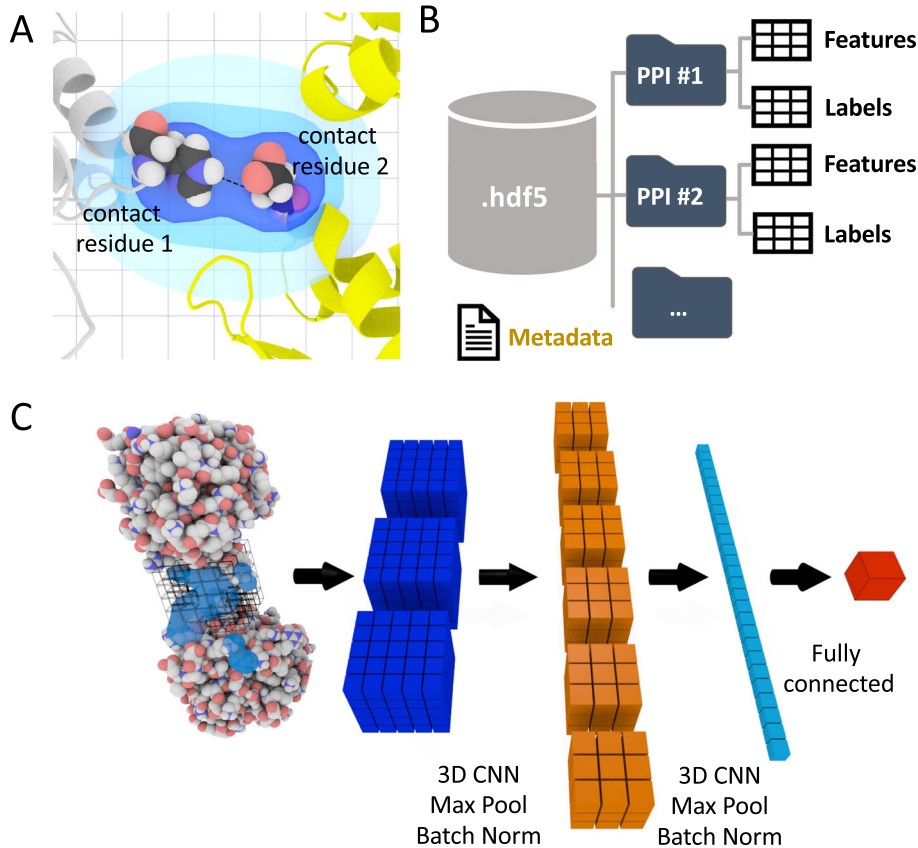

**Fig. 1 The DeepRank framework. A** The interface definition used by DeepRank. A residue is considered an interface residue if it is within a distance cutoff (5.5 Å by default, adjustable) of any atom on the other chain. The properties of interface residues or their atoms are used as features mapped on a 3D grid centered onto the interface. **B** Efficient storage of protein coordinates, features, and labels in HDF5 files. Given PDB files of protein-protein complexes, DeepRank determines interface residues, calculates features, and maps the features onto 3D grids, storing these data, along with necessary metadata into HDF5 files. These files are the input for the subsequent CNN (Convolutional Network). This HDF5 format greatly facilitates and speeds up the retrieval of specific information. **C** Illustration of the training process. Users may select a subset of features and/or a subset of PPIs (protein-protein interfaces) contained in the HDF5 file and use them as input for a 3D CNN. The example network consists of several layers that mix convolution, max pooling, batch norm operations as well as fully connected layers. The output of the network is the prediction of user-defined targets. Both classification and regression are supported.

**Table 1 Interface features predefined in DeepRank.**

| Feature Type | Feature Name | Details |
|---|---|---|
| Atom-level | Atom density | Density per element type[40] |
| | Atomic charges | Based on the OPLS force field[41] implemented in HADDOCK[19] |
| | Intermolecular electrostatic energy | Based on the OPLS force field[41] implemented in HADDOCK[19] |
| | Intermolecular van der Waals energy | Based on the OPLS force field implemented in HADDOCK[19] |
| Residue-level | Number of residue-residue contacts | Classified based on residue types[17] |
| | Buried Surface Area[42] | |
| | Position specific scoring matrix (PSSM)[43] | The log likelihood of 20 residue types appears at the specific position in a multiple sequence alignment |

3. Flexible target value definitions and calculations. Users may easily define problem-specific target values for their protein structures. For the scenario of computational docking, standard metrics to evaluate the quality of a docking model, i.e., by comparison to a reference structure, as used in the CAPRI (Critical Assessment of PRedicted Interactions)[23], are integrated into DeepRank. These include ligand RMSD (Root Mean Square Deviation)[24], interface RMSD (iRMSD)[24], FNAT (Fraction of Native Contacts)[24], CAPRI quality labels[24], and DockQ score[25]. DeepRank leverages pdb2sql[22] to perform these calculations efficiently.

4. Efficient data storage in HDF5 format. Dealing with sometimes tens of millions of small-size PDB files with rich feature representations presents a challenge both for the file system and for efficient training of deep neural networks. DeepRank stores the feature grids in HDF5 format, which is especially suited for storing and streaming very large and heterogeneous datasets. The general structure of the HDF5 generated by DeepRank is represented in Fig. 1B.

To train the neural network, DeepRank relies on the popular deep learning framework PyTorch[26]. The general network

architecture used in this work is illustrated in Fig. 1C. Starting from the HDF5 files, users can easily select which features and target value to use during training and which PPIs to include in the training, validation, and test sets. It is also possible to filter the PPIs based on their target values, for example by only using docking models with an iRMSD values above or below a certain threshold, thus discarding unrealistic data points. The input data are fed into a series of 3D convolutional layers, max pool layers, and batch normalization layers, usually followed by fully connected layers. The exact architecture of the network as well as all other hyper parameters can be easily modified by users to tune the training for their particular applications (see Supplementary Notes 1 and 4). The result of the training is stored in a dedicated HDF5 file for subsequent analysis.

**Application 1: Detecting crystal artifacts**. X-Ray crystallography is one of the most important experimental approaches to determine 3D structures of protein complexes (it accounts for >80% of all deposited PDB entries). This experimental technique first requires the proteins to be crystallized and then exposed to X-rays to obtain their structures. When it comes to structures of complexes, the resulting crystals often contain multiple interfaces, some of which are biologically relevant and some are mere artifacts of the crystallization process, the so-called "crystal interfaces" (Fig. 2A, B). Distinguishing crystal interfaces from biological ones, when no additional information is available, is still challenging. Several computational approaches have been proposed to distinguish such interfaces, among which PISA[18] and PRODIGY-crystal[16,17] show the highest prediction performances. PISA is based on six physicochemical properties: Free energy of formation, solvation energy gain, interface area, hydrogen bonds, salt-bridge across the interface, and hydrophobic specificity. PRODIGY-crystal is a random forest classifier based on structural properties of interfacial residues and their contacts[16].

We applied DeepRank to the problem of classifying biological vs. crystal interfaces. We trained and validated the 3D CNN specified in Supplementary Note 3 on the MANY dataset[27],

which consists of 2828 biological interfaces and 2911 crystal ones, only using Position Specific Scoring Matrix (PSSM) features. Each structure was first augmented by random rotation (30 times) before training. Early stopping on the validation loss was used to determine the optimal model (see Supplementary Fig. 3). The trained network was tested on the DC dataset[28], containing 80 biological and 81 crystal interfaces. On this test set, the trained network correctly classified 66 out of 80 biological interfaces and 72 out of 81 crystal interfaces (Fig. 2C). DeepRank thus achieved an accuracy of 86%, outperforming PRODIGY-crystal and PISA, which reported 74 and 79%, respectively[16] (Fig. 2D). While 89 test cases present at least one homolog in the MANY dataset, removing these cases from the testing dataset still leads to satisfying performance with an accuracy of 82%. (Supplementary Table 1).

**Application 2: Ranking docking models**. Computational docking is a valuable tool for generating possible 3D models of protein complexes and provides a complementary alternative to experimental structure determination. Given the 3D structures of individual proteins, docking aims at modeling their interaction mode by generating typically tens of thousands of candidate conformations (models). Those models are ranked using a scoring function to select the correct (near-native) ones (Fig. 3A). Although much effort is dedicated to improve the scoring[23,29–31], reliably distinguishing a native-like model from the vast number of incorrectly docked models (wrong models) remains a major challenge in docking.

We used HADDOCK[19] to generate a set of docking models of various qualities for the docking benchmark v5 (BM5) set[32], including both rigid-body docking, flexible docking, and final refined docking models. In this work, we focused on 142 dimers for which near-native models were available in the generated data sets, excluding all antibody-antigen complexes.

We trained the 3D CNN (architecture specified in Supplementary Note 3) using all available atomic and residue-based features (Table 1), mapped onto a grid of 30 × 30 × 30 Å³ with 1 Å

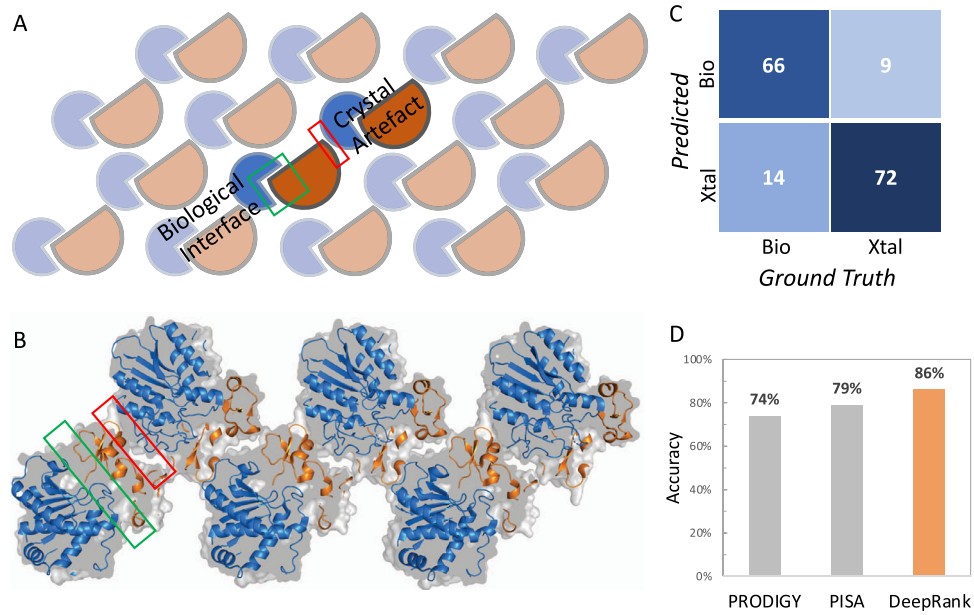

**Fig. 2 Classification of biological and crystal interfaces using DeepRank. A**, **B**. Illustration of the two types of interfaces, i.e., biological and crystal interfaces, found in a crystal. Protein molecules are orderly arranged in repetitive crystal units. Crystallographic interfaces can originate from the seeming interaction from the two neighboring crystal units, which may or may not represent biological interactions. The crystal structure shown in B corresponds to PDB entry 1ZLH. **C** Confusion matrix of DeepRank on the DC test dataset. **D** Accuracy of DeepRank, PISA, and PRODIGY-crystal on the DC dataset.

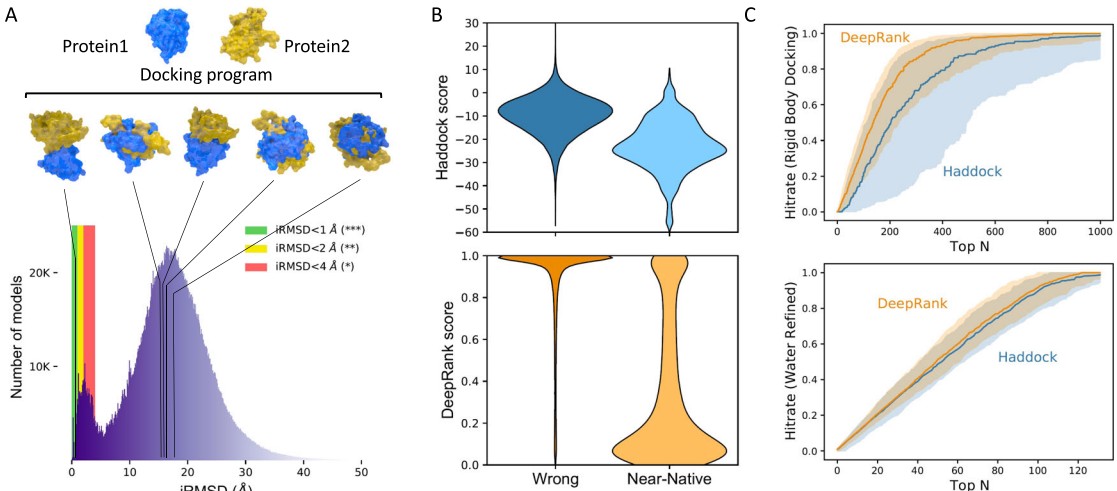

**Fig. 3 DeepRank applied to the docking scoring problem. A** Top: Using a docking software (e.g., HADDOCK[36]) a large number of docking poses between two proteins (here PDB ID: 1AK4) are generated. Bottom: Distribution of iRMSD (interface root mean squared deviation) values obtained for the docking benchmark 5[32] using HADDOCK in five docking scenarios (see Methods). Most conformations (93%) are wrong with iRMSD values larger than 4 Å and only less than 1% of the conformations have iRMSD values lower than 1 Å. **B** Distribution of the DeepRank and HADDOCK scores for the wrong models and near-native models from the rigid body docking stage. The lower the score the higher likelihood a model is predicted to be a near-native model. **C** Performance of DeepRank on the BM5 set compared to HADDOCK's scoring function. This data represents the predictions of both methods on 140 distinct test cases considered during the 10-fold cross-validation. Each individual test case contains about 3000 conformations of a single complex. The thick line marks the median of the Hit Rates (see Methods) calculated for individual test cases and the shaded area marks the 25–75% quantile interval. The data is shown up to the top 1000 (see Supplementary Fig. 6 for the full-range plot). Top: Rigid-body docking models only; Bottom: Water refined models only. HADDOCK uses different scoring functions for models generated in different stages: rigid-body, flexible-docking, and water-refinement stages (see Methods).

resolution. The network was trained on over 300,000 labeled docking conformations to classify models as near-native or wrong. The DeepRank score, i.e., the predicted likelihood of a model to be a wrong model, was then used to rank the models for each case. To ensure objective evaluations, we conducted 10-fold cross-validation at the level of complexes, i.e., all models for a given complex are used exclusively in either training, validation, or test sets in each fold.

DeepRank performs well on HADDOCK models generated in the rigid-body docking stage. The DeepRank scores are well separated between near-native and wrong models (Fig. 3B), while the HADDOCK scores present a significant overlap between those two classes. In terms of ranking performance, DeepRank outperforms HADDOCK on rigid-body docking models by selecting more near-native models among Top N (Fig. 3C top). The narrow spread of the values obtained with DeepRank, illustrated by the 25–75% quantile interval, indicates that DeepRank is rather consistent in its ranking of different cases, while HADDOCK presents poor performance for some cases. This difference might be explained by the fact that DeepRank is less sensitive to the detailed energetics of the interface than the HADDOCK score.

The differences between DeepRank and HADDOCK are less pronounced for water-refined docking models (Fig. 3C bottom). However, note that HADDOCK requires using different scoring functions for models generated in rigid-body, flexible-docking, and water-refinement stage while DeepRank use the same scoring function for all stages (see Methods and Supplementary Fig. 6). An analysis of Success Rate of DeepRank and HADDOCK at the different stages (Supplementary Fig. 7) confirms the good performance of DeepRank that slightly outperforms HADDOCK for each model type. This confirms again the robustness of the DeepRank score, since it provides a single score that performs well across differently refined models.

To further test the performance of DeepRank we have trained a final 3D CNN model using the docking conformations of all the 142 BM5 dimer complexes and applied it to 13 cases from the CAPRI score set[33]. We compared DeepRank with three leading scoring functions, the HADDOCK[19] scoring function that uses an energy-based approach; the recently developed iScore[20,21] a graph-kernel based scoring function; and DOVE[5] a recent deep-learning method also based on 3D CNNs. DeepRank is competitive with these scoring functions, even outperforming them on some cases (Supplementary Fig. 8 and Supplementary Table 2). Our results also suggest the ability of DeepRank to correctly identify favorable interactions that are ignored by the other methods, which might indicate a possible complementarity of these approaches (Supplementary Figs. 9 and 10).

## Discussion

We have presented here our DeepRank framework, demonstrating its use and performance on two structural biology challenges. Its main advantages are as follows:

1. From a user's perspective, the platform provides a user-friendly interface. It implements many options that can be easily tuned. It provides flexibility through the featurization and the design of the neural network architecture (see code snippets in Supplementary Note 4). This makes it directly applicable for a range of problems that use protein–protein interfaces as input information.
2. From a developer's perspective, DeepRank is developed as a software package following software development standards[34] including version control, continuous integration, documentation, and easy addition of new features. This flexibility increases the maintainability and further development of DeepRank by the community, for example, to allow predicting mutation effects on single protein structures.

3. Computational efficiency: in all stages, DeepRank has been developed to make it possible to use millions of PDB files to train models, and test their performance.

4. Finally, the performances competing and outperforming the state-of-the-art on two different research problems demonstrate the versatility of DeepRank in general structural biology.

When applied to the classification of biological versus crystallographic interfaces (application 1), the trained network (provided in Data Availability) shows satisfying performance leading to a better classification than competing methods, PRODIGY-crystal and PISA. This improvement is due to the use of evolution information through the PSSM and from the use of deep neural network that are capable of learning the subtle differences between the interaction patterns of the two types of interfaces.

When considering the scoring problem (application 2), DeepRank performs especially well on scoring rigid-body HAD-DOCK models, outperforming the HADDOCK's rigid-body-specific Score (Fig. 3C). Since rigid-body docking is the first modeling stage of HADDOCK, incorporating DeepRank scoring in the HADDOCK docking workflow is expected to improve the modeling success rate by passing more high-quality models to the later flexible refinement stages. This result also indicates that our trained network (provided in Data Availability) could be generally applicable to models from a variety of rigid-body docking software. DeepRank is robust on different type of models (rigid-body, flexible-refined, water-refined) (Fig. 3C) and not sensitive to clashes at the interface. This wide applicability range is important in experiments like the community-wide CAPRI scoring experiment where a mixture of highly refined and rigid-body models (that often present unphysical atomic arrangements, or clashes) have to be scored[23]. While DeepRank does not significantly outperform HADDOCK iScore nor DOVE on selected cases from previous rounds of CAPRI, it does identify different conformations as being near-native (Supplementary Fig. 9), indicating a complementarity in scoring. The comparison of the different methods clearly illustrates the difficulty in obtaining a model that performs consistently across the diversity of PPIs and calls for more research to engineer better featurization, datasets, and scoring functions.

Currently DeepRank converts irregular-shaped PPIs into structured data (i.e., 3D grids) so that 3D CNN can be applied. These structured 3D grids could also be used with equivariant neural networks[35] that naturally incorporate translation- and rotation-invariance and hence avoids the data augmentation that is sometimes needed when using 3D CNN. The use of non-structured geometric data such as graphs[7], surfaces[6], or point clouds as input, offer additional opportunities for the future development of DeepRank. For example, MaSIF[6] exploits geodesic CNN to extract protein interaction fingerprint and therefore only requires data augmentation in 2D instead of 3D. However, the data preprocessing required by MaSIF to determine protein surface patches, calculate polar coordinates and map the features, is about 48 times more computationally demanding and 7 times more memory demanding than computing all the 3D grids required by DeepRank (see Supplementary Table 3). This hinders the applicability of MaSIF to large-scale analyses on millions of protein models obtained for example in computational docking or large-scale modeling of mutations. Nevertheless, considering the potential of geometric learning with respect to rotation-invariance, it would be useful to extend DeepRank with geometric deep learning techniques to more efficiently represent PPIs with highly irregular shapes. Another enhancement would be to extend the framework to handle complexes containing more than two chains to broaden its application scope.

In summary, we have described an open-source, generic, and extensible deep learning framework for data mining very large datasets of protein-protein interfaces. We demonstrated the effectiveness and readiness of applying DeepRank on two different challenges in structural biology. We expect DeepRank to speed-up scientific research related to protein interfaces by facilitating the tedious steps of data preprocessing and reducing daunting computational costs that may be associated with large-scale data analysis. Its modularized and extendable framework bears great potential for stimulating collaborative developments by the computational structural biology community on other protein structure-related topics and will contribute to the adoption and development of deep learning techniques in structural biology research.

## Methods

**Gaussian mapping of the atomic and residue features**. The atomic and residue features are mapped on a 3D grid using Gaussian functions. The contribution $w_k$ of atom $k$ (or residue) to a given grid point follows a hence a Gaussian distance dependence:

$$w_k(r) = v_k \exp(||r - r_k||^2/2\sigma^2) \tag{1}$$

where $v_k$ is the value of the feature, $r_k$ the (x,y,z) coordinate of atom $k$, and $r$ the position of the grid point. For atomic features the standard deviation $\sigma$ is taken as the van der Waals radius of the corresponding atom. In case of residue-based features, $r_k$ is taken as the position of the alpha-carbon atom of the residue and $\sigma$ as the van der Waals radius of that atom.

For pairwise features, such as interaction energy terms, the value of $v_k$ is defined as the sum of the all the interaction terms between atom (residue) k and its contact atoms(residues): $v_k = \sum_l v_{kl}$ where $v_{kl}$ is the interaction term between atoms(residues) $k$ and $l$.

**Application 1: Detecting crystal artifacts**. Data Sets: The original MANY dataset[27] consists of 2831 biological interfaces and 2912 crystal ones. Entries 2uuy_1, 2p06_1 and 3dt5_1 were excluded due to all zero values of PSSM information content (IC) and 1gc2_5 was excluded due to failed residue density calculation. We thus used 2828 biological interfaces and 2911 crystal ones, which was used to create the training (80%) and validation (20%) sets while maintaining the balance between positive and negative data. The training set was then further augmented by randomly rotating each complex 30 times. The DC dataset[28] was used as a test set. It contains 80 biological and 81 crystal interfaces, excluding 3jrz_1 due to failed residue density calculation. All entries of both the MANY and the DC dataset have been previously refined[16] using the refinement protocol of our HADDOCK web server[36].

Training and evaluations: The architecture of the CNN used in our experiments is shown in Supplementary Fig. 2. We used the cross-entropy and log softmax as loss and scoring functions, respectively, and stochastic gradient descent as optimizer. The learning rate was set to 0.0005, momentum to 0.9, weight decay to 0.001, and batch size to 8. As features we used only the PSSM as evolutionary information, which has been demonstrated to be useful for charactering protein interfaces[4]. We used $10 \times 10 \times 10$ Å³ grids with a resolution of 3 Å. This low-resolution grid is sufficient to capture the variation of the PSSM which are residue-based features and thus less finely defined than atomic-based features.

**Application 2: Ranking docking models**. Data sets: The dataset for scoring was generated using HADDOCK, an integrative modeling software[19]. Docking in HADDOCK follows three stages: (1) it0: rigid-body docking, (2) it1: semi-flexible refinement by simulated annealing in torsion angle space, and (3) itw: final refinement by short molecular dynamics in explicit solvent (default: water). We used HADDOCK[37] to systematically generate a set of docking models for the docking benchmark v5 set[32], BM5. In order to generate a suitable amount of near-native models both guided and ab-initio dockings were performed following five different scenarios: (1) Refinement of the bound complexes (50/50/50 models for it0/it1/water stages, referred to as "refb"), (2) guided docking using true interface defined at 3.9 Å cutoff as ambiguous interaction restraints (1000/400/400 models for it0/it1/water, "ti"), (3) guided docking using true interface defined at 5.0 Å cutoff (1000/400/400 models for it0/it1/water, "ti5"), (4) ab-initio docking with center of mass restraints (10,000/400/400 models for it0/it1/water, "cm"), and (5) ab-initio docking with random surface patch restraints (10,000/400/400 models for it0/it1/water, "ranair"). Scenarios 2–5 are unbound-unbound docking, providing real-life challenges (i.e., conformational changes upon binding) to DeepRank. BM5 consists of 232 non-redundant cases (the non-redundancy was here evaluated at the SCOP family level[32]). In total, we generated ~5.84 million HADDOCK models (25,300 models per case) corresponding to 11 TB of data (3D coordinates plus feature grids) in HDF5 format. In this study, we focused on 142 dimers, i.e., cases

with two chains (1IRA was excluded due to the lack of PSSM for the short chain A).

As the models were not clustered, a large degree of redundancy exists in the dataset, with very similar conformations being represented multiple times. After experimentation, we observed that considering only a subset of about 420 K models (~3000 models per complex) was sufficient to accurately represents the available data (see Supplementary Fig. 4 for details). About 30 K models were near-native models among these 420 K models. Each docking model was randomly rotated to limit the sensitivity of our results to a particular orientation of the PPIs.

Network architectures and weight optimizations: The architecture of the CNN in our experiment is shown in Supplementary Fig. 5. We used the cross-entropy loss function over the raw scoring output. We used the Adam optimizer with a learning rate of 0.001. The batch size during training was 100. We used the full set of 36 physico-chemical features (channels) that are predefined in DeepRank (Table 1). We used a grid size was 30 x 30 x 30 Å$^3$ with a resolution of 1 Å to accurately capture the variation of atomic-based features and to adequately resolve the average buried surface area per monomer (BSA) of a dimer PPI in the BM5 set, that is about 909 Å$^2$. We evaluate the BSA as:

$$BSA = \frac{1}{2}(ASA_{chainA} + ASA_{chainB} - ASA_{complex})\quad(2)$$

where ASA stands for the accessible surface area.

Because our dataset is highly imbalanced (only around 7.1% of the models are positives, i.e., with iRMSD 4 Å), for training the network we assigned class weights to the loss function, proportional to the class sizes, that is, the class weights are 0.071 and 0.929 for the negatives and positives, respectively so that the loss function penalizes equally errors made on the positive and on the negative dataset.

Training, validation, and testing: We performed a 10-fold cross-validation on the BM5 set. In every fold, out of 142 complexes, we used models from 114 complexes for training, 14 complexes for validation, and 14 complexes were reserved for testing. In this way, in total 140 out of the 142 complexes were used in the test sets (two complexes BAAD and 3F1P are not used in the testing to keep the testing set having an equal number of complexes for each fold). The training, validation, and testing sets are disjunctive at complex-level so that complexes in the testing set are never seen by the neural networks during its training and validation. In the end, we trained one network for each fold (thus in total 10 trained networks for 10-CV), and the set-aside independent test set was evaluated on this network.

The network is trained to perform positive/negative classification of models, outputting two scores as predicted probability of an input model being near-native or wrong. We then ranked the docking models based on their predicted scores of being wrong models to be consistent with widely-used energy-based scoring functions: i.e., a low prediction score indicates that the function predicts this model is of high quality. For the final analysis (Supplementary Fig. 7), we merged prediction results from 10 folds in our performance reports on the BM5 set.

To evaluate DeepRank on the CAPRI score set, we have first trained one network on all the 449,158 docking models from 142 BM5 complexes. This trained neural network model was then used on the CAPRI score set[33].

HADDOCK scoring functions: HADDOCK uses three different scoring functions for models generated at different docking stages. We neglected the AIR restraints to facilitate the comparison between the different model qualities. We used the following functions:

$$HADDOCK_{it0} = 0.01E_{vdw} + 1.0E_{elec} + 1.0E_{desol} - 0.01BSA\quad(3)$$

$$HADDOCK_{it1} = 1.0E_{vdw} + 1.0E_{elec} + 1.0E_{desol} - 0.01BSA\quad(4)$$

$$HADDOCK_{itw} = 1.0E_{vdw} + 0.2E_{elec} + 1.0E_{desol}\quad(5)$$

where HADDOCK$_{it0}$, HADDOCK$_{it1}$, and HADDOCK$_{itw}$ are the scoring functions used respectively for rigid-body docking, semi-flexible docking and water-refined models. E$_{vdw}$ is the van der Waals intermolecular energy, E$_{elec}$ the electrostatic intermolecular energy, E$_{desol}$ the desolvation energy and BSA the buried surface area.

Evaluation metrics: Hit Rate and Success Rate are used to evaluate the performance of scoring functions. The Hit Rate is defined as the percentage of hits (models with iRMSD ≤4 Å) in the top ranked models for a specific complex:

$$Hit\,Rate(K) = \frac{n_{hits}(K)}{M}\quad(6)$$

where $n_{hits}(K)$ is the number of hits (i.e., near-native models) among top models and $M$ the total number of near-native models for this case. The Hit Rate was calculated for each individual case in our test set and statistics across the different cases (median, 1$^{st}$ and 3$^{rd}$ quartile) were calculated. As the total number of models varies between cases, these statistical values were only evaluated from $K = 1$ to $K = N_{min}$ where $N_{min}$ is the smallest number of models that all cases have. The Success Rate shown in Supplementary Fig. 7 is defined as the percentage of complexes for which at least one near-native model is found in the top N selected models. It is therefore defined as:

$$Success\,Rate = \frac{n_{successful\_cases}(K)}{N}\quad(7)$$

where $n_{successful\_cases}(K)$ is the number of cases with at least one near-native model among top models, and $N$ is the total number of cases.

**Reporting summary**. Further information on research design is available in the Nature Research Reporting Summary linked to this article.

## Data availability
The PDB files and PSSM files for the two experiments along with the code for training and post analysis have been deposited in SBGrid (data.sbgrid.org, https://doi.org/10.15785/SBGRID/843).

## Code availability
The DeepRank software has been released to the Python Package Index at https://pypi.org/project/deeprank/ (https://doi.org/10.5281/zenodo.3735042)[38]. Its source code and documentation are freely available at https://github.com/DeepRank/deeprank and https://deeprank.readthedocs.io, respectively. The PSSMs used in this paper were calculated using our PSSMgen package: https://github.com/DeepRank/PSSMGen (https://doi.org/10.5281/zenodo.4509544)[39].

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

## Acknowledgements

The project is supported by the ASDI grant provided by Netherlands eScience Center (Grant number ASDI.2016.043), by SURF Open Lab "Machine learning enhanced HPC applications" grant (AB/FA/10573), and by a "Computing Time on National Computer Facilities" grant (2018/ENW/00485366) from NWO (Netherlands Organization for Scientific Research). AMJJB acknowledges financial support from the European Union Horizon 2020 projects BioExcel (675728, 823830) and EOSC-hub (777536). This work was carried out on the Dutch national e-infrastructure with the support of SURF Cooperative (Grant ID: 2018/ENW/00485366). LX and DM acknowledge financial support by the Hypatia Fellowship from Radboudumc (Rv819.52706). We thank Dr. Valeriu Codreanu, Dr. Caspar van Leeuwen, and Dr. Damian Podareanu from SURFsara for providing HPC support for efficient data processing using Cartesius, the Dutch national supercomputer.

## Author contributions

L.X. and A.M.J.J.B. designed and supervised the project. L.X. contributed to the development and evaluations of DeepRank. A.M.J.J.B. generated the HADDOCK docking models. C.G., L.X., S.G. and N.R. designed and developed the software. C.G. and F.A. performed the experiments for Application 1 (classification of biological interfaces vs. crystal ones). S.G. performed the experiments for Application 2 (ranking docking models) and contributed to experiments in Application 1. D.M. contributed to writing the documentation and evaluations of DeepRank. MR contributed to the development of the software and the evaluation of the computational efficiency of DeepRank and MaSIF. L.R. organizes the project and contributed to the discussions and development of DeepRank. All authors contributed to the manuscript writing.

## Competing interests

The authors declare no competing interests.
