## [Peer Review File · Nature Communications]

DeepRank: A deep learning framework for data mining 3D protein-protein interfacesREVIEWER COMMENTS

Reviewer #1 (Remarks to the Author):

The paper titled "DeepRank: A deep learning framework for data mining 3D protein protein interfaces" presents a deep learning framework, called DeepRank, for data mining of PPIs using 3D convolutional neural networks (CNN). The training is done using features of PPIs mapped onto 3D grids with a user-specified CNN. The performance of DeepRank was tested on the classification of biological versus crystallographic PPIs and the ranking of docking models. Overall the method performs very well. My comments are below:

1. the authors have used a basic CNN with layers, which is a proven architecture, but it could potentially have been better if they tried some kind of RNN architecture in combination with the CNN, because of the sequential nature of the dataset. Of course it is up to the authors to try different architectures but I wonder if they have actually tried it and did not mention it in the paper because of poor performance. This would be interesting for the readers.
2. There were some strong technical points which should be appreciated such as using MPI for featurization. This is a good parallelisation technique to save time and it should be appreciated.
3. The last paragraph of the Introduction section reads as "In both challenges, DeepRank performs competitively or outperforms state-of-the-art approaches, showing its applicability and potential for structural biology". It will increase the readability to list all the predictors DeepRank is benchmarked against and their performance in a table (such as PRODIGY-crystal, PISA) rather than only on a figure (figure 2).
4. It is common knowledge that the train and test sets have to be independent of each other to estimate the unbiased performance of the model. When the authors select their train, validation (80%, 20%) and test sets, surely there are no same protein present in the train (80% + 20%) and validation set. Apart from that separation, did they consider any other form of separation such as the sequence identity between and among the proteins of these datasets? If they did, it will be a valuable piece of information for readers.
5. For the application 2, It will be interesting to gauge the performance of the model with an evaluation matrix that is sensitive to data imbalance problem, such as Matthews correlation coefficient (MCC), in-addition to gauging the performance by the accuracy (even though the accuracy is fixed by assigning class weights to the loss function).
6. The sentence "DeepRank stores the feature grids in HDF5 format" is necessary, however, the description of HDF5 format and its advantages can go into the supplement as it is an implementation detail, not part of the prediction algorithm.
7. According to the supplementary material, the classification of biological vs crystal interfaces problem uses a neural network trained only after two epochs of training. According to Fig S3, the performance of the model trained for two epochs does not perform significantly better compared to the model before training with initialised parameters. It will be interesting if the authors explain the reason for this in the paper.

Reviewer #2 (Remarks to the Author):

The paper addresses an important problem of structural modeling of protein-protein interactions. The progress in computing hardware and new methods of Deep Learning that take advantage of it, open new opportunities to increase accuracy and reliability of such modeling. Recent advances in prediction of individual protein structures at CASP14, based on the Deep Learning have not yet

translated to protein assemblies. Thus, development of adequate Deep Learning techniques for protein docking is high on the current agenda among docking researchers.

The authors put forward an open-source Deep Learning framework for assessment of protein-protein interfaces. The details of the 3D CNN and its application are properly described and illustrated. The procedure was trained on a set of interfaces and applies to two important problems: (1) discrimination of protein-protein crystal contacts and (2) scoring of the docking predictions. In both applications, the approach was compared with existing state-of-the-art procedures, showing equal or better performance.

The developed framework demonstrates the utility of the Deep Learning in modeling of protein assemblies and will be a useful tool for the research community. Thus, I recommend publishing it in Nature Communications.

Reviewer #3 (Remarks to the Author):

General Comments:

Renaud et al. reported DeepRank, a deep learning-based framework and software platform, for data mining of structural information for protein-protein interactions. Specifically, a 3D CNN-based deep learning model was involved to learn from the discretized 3D structures of protein-protein interfaces for various types of supervised learning tasks. The two tasks showcased to benchmark the model behaviors include binary classification of biological versus crystal contacts and scoring structure models from protein docking.

From the perspective of an enabling platform that the authors are targeting, DeepRank was efficient in the 3D data processing (Table S2), partly thanks to the use of HDF5 data format. Its deep learning portion relies on pyTorch, a popular deep learning library.

From the perspective of advancing PPI modeling, DeepRank was compared to competing methods in aforementioned tasks. Especially in the second task of ranking structure models from docking, comprehensive experiments were performed on various docking scenarios/stages and various learning scenarios.

Please find specific comments below that could improve the manuscript.

Major:

1. Since one of the targets is an enabling platform, the authors may want to include some materials about the flexibility of DeepRank, which was mostly discussed in data generation but not in deep learning. The authors can expand Description-B in main text and/or Sec. 1B in the supplement by discussing about the flexibility of constructing and training various 3D CNN models. Examples include various architecture hyperparameters among other model flexibility as well as training loss (for instance if regression rather than binary classification is needed for a different task) among other task flexibility. It is understandable that deep learning in DeepRank relies on pyTorch and many such flexibility comes from pyTorch. But such model/task flexibility is important to be pointed out for the broader audience.

2. Application 1 (biological vs crystal PPIs). The DC test set was claimed to be "independent" from the MANY training set. To rigorously assess and communicate DeepRank's generalizability, it is important to know how "different" the test set was to the training. The similarity can be measured by the maximum interface similarity for each DC test case compared to the training set (max over all training examples). And the performances can be reported separately for test cases in individual similarity ranges/bins.

3. Application 2 (scoring PPI models). The use of CAPRI Score test set is commendable - it is posing a real-world challenge to model generalizability where training examples and test cases were very likely of different distributions as they were generated by different protein docking protocols.

One suggestion here is to include a comparison to Wang et al. (ref. 5), a 3D CNN model, for the same application. The reason is that only HADDOCK score and iScore were compared whereas neither is a deep learning based method. The goal to compare with Wang et al. is not to show that DeepRank had to beat every method. Rather, the goal is to show that DeepRank in the current setting is competitive enough with the state of the art while providing this enabling platform with efficiency and flexibility.

Minor:

1) As the platform is broader than ranking (PPI models) and is generic in extracting 3D structure information from protein-protein interfaces, the current name of DeepRank may be limiting a broader impact. The platform is more about data mining from 3D PPI than that for PPI model ranking.

2) Although GNNs and equivariant NNs were recognized in the introduction or discussion, their latest developments directly for application 2 (scoring PPI models) were not. Examples include

GNN for scoring PPIs: <https://doi.org/10.1002/prot.25888>

Equivariant NNs for scoring PPIs: <https://doi.org/10.1002/prot.26033>

3) Please consider reporting memory comparison in the supp. 3-E in addition to time comparison.

4) The DeepRank data processing time in Table S2 was on average 5 seconds only. As multiple sequence alignment (MSA) costs much time and is needed for PSSM features, I wonder whether such processing time did not include that for MSA. Please clarify. If MSA is additionally part of the data processing in DeepRank, please consider including more details (library, software, parameters etc).

Rebuttal to the reviewer's comments for the publication entitled : *DeepRank: A deep learning framework for data mining 3D protein-protein interfaces*

We would like to address our gratitude for all reviewer's comments and for taking the time to review and assess the quality of our work. We have done our best to address all the comments and suggestions and have detailed our answers below. We have also indicated potential changes in the text and / or supplementary material and highlighted those in the revised manuscript and supplementary material.

Comments from Reviewer #1

Question 1. The authors have used a basic CNN with layers, which is a proven architecture, but it could potentially have been better if they tried some kind of RNN architecture in combination with the CNN, because of the sequential nature of the dataset. Of course it is up to the authors to try different architectures but I wonder if they have actually tried it and did not mention it in the paper because of poor performance. This would be interesting for the readers.

Answer 1. Thank you for your comment. It is true that we do not exploit any information about the sequential nature of the protein data. Since we are directly exploiting the 3D atomic coordinates, residues that are far away in the sequence may be close to each other, and we are therefore more interested in the spatial information of interface residues. Thus, we opted for only CNN architecture. Combining sequential and spatial information is challenging but could be the focus of future developments of DeepRank.

Q2. There were some strong technical points which should be appreciated such as using MPI for featurization. This is a good parallelization technique to save time and it should be appreciated.

A2. Thank you for recognizing the technical strength of DeepRank. We hope that future users will take advantage of it and further improve our tool.

Q3. The last paragraph of the Introduction section reads as "In both challenges, DeepRank performs competitively or outperforms state-of-the-art approaches, showing its applicability and potential for structural biology". It will increase the readability to list all the predictors DeepRank is benchmarked against and their performance in a table (such as PRODIGY-crystal, PISA) rather than only on a figure (figure 2).

A3. Thank you for your suggestions to make our results more visible. To highlight the results presented in the paper we have summarized the performance of DeepRank at the end of the introduction by adding the following sentence :

"To demonstrate its applicability and potential for structural biology, we apply it to two different research challenges. We first present the performance of DeepRank for the classification of biological vs. crystallographic PPIs. With an accuracy of 86%, DeepRank outperforms the state-of-the-art methods PRODIGY-crystal and PISA that respectively reach an accuracy of 74% and 79%. We then present the performance of DeepRank for the scoring

of models of protein-protein complexes generated by computational docking. We show here that DeepRank is competitive and sometimes outperforms three state-of-the-art scoring functions: HADDOCK, iScore, and DOVE.”

Q4. It is common knowledge that the train and test sets have to be independent of each other to estimate the unbiased performance of the model. When the authors select their train, validation (80%, 20%) and test sets, surely there are no same protein present in the train (80% + 20%) and validation set. Apart from that separation, did they consider any other form of separation such as the sequence identity between and among the proteins of these datasets? If they did, it will be a valuable piece of information for readers.

A4. Thank you for your comments. In the case of Application 2 (i.e. the ranking of docked conformations), all cases in the BM5 dataset are non-redundant at the SCOP family level. We clarified this by adding the following sentence in the Methods section (Application 2 / Dataset):

“BM5 consists of 232 non-redundant cases (the non-redundancy is defined at the SCOP family level).”

For Application 1, we explored the redundancy of the DC test set and the impact it has on the performance of DeepRank. As detailed in 2C of the supplementary material, 89 of the 161 complexes from the DC dataset (i.e. our testing dataset) have at least 1 homolog in the MANY dataset (our training dataset).

However when removing these cases from the test dataset, DeepRank achieved an accuracy of 81.9% on this non-redundant DC set, that is comparable to the accuracy of 85.7% obtained on the entire DC set (**Table S1**). As PRODIGY-crystal and PISA were tested on the entire DC dataset, and PRODIGY was trained on MANY as well, we have left **Fig. 2D** as it was. We have added the following sentences in the main text to highlight the impact of redundancy on our results:

“While 89 test cases present at least one homolog in the MANY dataset, However, removing these cases from the testing data set still leads to satisfying performance with an accuracy of 82%. (Table S1).”

In addition, we added a new section in the supplementary material to present these results :

“2-C Redundancy of DC set and related DeepRank performance

To explore the redundancy of the DC test dataset, we calculated sequence identities of DC complexes against those from the MANY dataset on chain level. We used 30%, a standard cutoff for sequence identity to define a homolog. As a result, 89 out of the 161 complexes from the DC set have at least 1 homolog in the MANY set. These 89 complexes have each on average 4 homologs in the training set. This consequently leaves 72 complexes from the DC that do not have any homologs in the MANY set.

We have then evaluated the performance of the trained model on the 72 complexes of the DC dataset that do not present any homologs. From the table below, we can see that DeepRank achieved a comparable accuracy on DC complexes without homologs (81.9%) than on the entire datasets (85.7%). This therefore excludes the risk that the performance of our model is due to the presence of homologous sequences in the training and test data set.

Table S1. Performance on redundant and non-redundant DC complexes

	Accuracy	TP	FN	TN	FP	#complexes
DC complexes without homologs	81.9%	32	9	27	4	72
DC complexes with homologs	88.7%	34	5	45	5	89
All DC complexes	85.7%	66	14	72	9	161

Q5. For the application 2, It will be interesting to gauge the performance of the model with an evaluation matrix that is sensitive to data imbalance problem, such as Matthews correlation coefficient (MCC), in-addition to gauging the performance by the accuracy (even though the accuracy is fixed by assigning class weights to the loss function).

A5. We agree with the reviewer that the MCC is a useful metric for evaluating binary classification of imbalanced datasets. We have now implemented this metric and it is part of the DeepRank framework. In particular, MCC for DeepRank on BM5 equals 0.49.

However, we would like to clarify that accuracy was not used as a metric to gauge the performance of the model in application 2. We used the cross-entropy as our loss function for training the network and log softmax as a scoring function (see Methods). To deal with the data imbalance problem, we weighted the loss accordingly. In place of accuracy, the hit rate and success rate, which are standard evaluation metrics in docking scoring, were used to assess the performance of our model.

Q6. The sentence "DeepRank stores the feature grids in HDF5 format" is necessary, however, the description of HDF5 format and its advantages can go into the supplement as it is an implementation detail, not part of the prediction algorithm.

A6. Thank you for pointing that out. We agree with your comment and we have rephrased and moved the description of the HDF5 format to section 1A of supplementary material

"The HDF5 file format is increasingly popular for storing large dataset used in deep learning applications as it allows for efficient memory usage and fast input/output operations. In addition all the required metadata (code versioning, protocol generation and so on) can be directly included in the HDF5 file which partly aligns with the FAIR principle (Findable, Accessible, Interoperable and Reusable). "

Q7. According to the supplementary material, the classification of biological vs crystal interfaces problem uses a neural network trained only after two epochs of training. According to Fig S3, the performance of the model trained for two epochs does not perform significantly better compared to the model before training with initialized parameters. It will be interesting if the authors explain the reason for this in the paper.

A7. Thank you for noticing this issue in our figure. The first data point reported in **Fig. S3** corresponds to the loss *after* the first epoch and not the loss obtained with the initial values of the network's parameters. To clarify that, we have replotted **Fig. S3** so that the first data point appears at epoch=1 instead of epoch=0 as it was the case before. We have also updated the text of the section 2B of the supplementary material that now reads :

“As the validation loss is lowest at epoch 2, we used the trained network at epoch 2 as our final network.”

Reviewer #2 (Remarks to the Author):

The paper addresses an important problem of structural modeling of protein-protein interactions. The progress in computing hardware and new methods of Deep Learning that take advantage of it, open new opportunities to increase accuracy and reliability of such modeling. Recent advances in prediction of individual protein structures at CASP14, based on the Deep Learning have not yet translated to protein assemblies. Thus, development of adequate Deep Learning techniques for protein docking is high on the current agenda among docking researchers.

The authors put forward an open-source Deep Learning framework for assessment of protein-protein interfaces. The details of the 3D CNN and its application are properly described and illustrated. The procedure was trained on a set of interfaces and applies to two important problems: (1) discrimination of protein-protein crystal contacts and (2) scoring of the docking predictions. In both applications, the approach was compared with existing state-of-the-art procedures, showing equal or better performance.

The developed framework demonstrates the utility of the Deep Learning in modeling of protein assemblies and will be a useful tool for the research community. Thus, I recommend publishing it in Nature Communications.

Answer to reviewer #2 : We would like to express our gratitude to reviewer 2 for his positive comments and for supporting our work. We hope that many users will share the same sentiment and will adopt and further extend DeepRank.

Reviewer #3 (Remarks to the Author):

General Comments:

Renaud et al. reported DeepRank, a deep learning-based framework and software platform, for data mining of structural information for protein-protein interactions. Specifically, a 3D CNN-based deep learning model was involved to learn from the discretized 3D structures of protein-protein interfaces for various types of supervised learning tasks. The two tasks showcased to benchmark the model behaviors include binary classification of biological versus crystal contacts and scoring structure models from protein docking.

From the perspective of an enabling platform that the authors are targeting, DeepRank was efficient in the 3D data processing (Table S2), partly thanks to the use of HDF5 data format. Its deep learning portion relies on pyTorch, a popular deep learning library.

From the perspective of advancing PPI modeling, DeepRank was compared to competing methods in aforementioned tasks. Especially in the second task of ranking structure models from docking, comprehensive experiments were performed on various docking scenarios/stages and various learning scenarios.

Question 1. Since one of the targets is an enabling platform, the authors may want to include some materials about the flexibility of DeepRank, which was mostly discussed in data generation but not in deep learning. The authors can expand Description-B in main text and/or Sec. 1B in the supplement by discussing about the flexibility of constructing and training various 3D CNN models. Examples include various architecture hyperparameters among other model flexibility as well as training loss (for instance if regression rather than binary classification is needed for a different task) among other task flexibility. It is understandable that deep learning in DeepRank relies on pyTorch and many such flexibility comes from pyTorch. But such model/task flexibility is important to be pointed out for the broader audience.

Answer 1. Thank you for your suggestion. We have added a section to the Supplementary Material (i.e. Section 4 – Usage and flexibility of DeepRank) to illustrate the use of DeepRank and highlight its flexibility. We demonstrate in that section how to generate the data, define a neural network architecture, fix all the hyperparameters and train the model. We also pointed the readers to our comprehensive documentation online that contains many tutorials as well as the comprehensive API of the software: <https://deeprank.readthedocs.io/>

Furthermore, we have added a sentence to the main text that points at this new section of the Supplementary Material:

“The exact architecture of the network as well as all other hyper parameters can be easily modified by users to tune the training for their particular applications (see Supplementary Materials: Sections 1 and 4).”

We hope that with these additions readers will understand how to use and adapt DeepRank for their specific applications.

Q2. Application 1 (biological vs crystal PPIs). The DC test set was claimed to be "independent" from the MANY training set. To rigorously assess and communicate DeepRank's generalizability, it is important to know how "different" the test set was to the training. The similarity can be measured by the maximum interface similarity for each DC test case compared to the training set (max over all training examples). And the performances can be reported separately for test cases in individual similarity ranges/bins.

A2. Thank you for pointing this out. Reviewer #1 asked the same question (Question4) and we have provided a detailed answer there. Please refer to our detailed answer there.

Q3. Application 2 (scoring PPI models). The use of CAPRI Score test set is commendable - it is posing a real-world challenge to model generalizability where training examples and test cases were very likely of different distributions as they were generated by different protein docking protocols.

One suggestion here is to include a comparison to Wang et al. (ref. 5), a 3D CNN model, for the same application. The reason is that only HADDOCK score and iScore were compared whereas neither is a deep learning based method. The goal to compare with Wang et al. is not to show that DeepRank had to beat every method. Rather, the goal is to show that DeepRank in the current setting is competitive enough with the state of the art while providing this enabling platform with efficiency and flexibility.

A3. Thank you for encouraging us to compare our results with more methods. Following your advice we have applied DOVE to the 13 CAPRI cases we have used to benchmark the different methods. We have updated **Fig. S8** and **Table S2** to report the hit rate we have obtained with DOVE on the different complexes. As seen in this figure, DeepRank is competitive with DOVE on many complexes, sometimes outperforming it, sometimes not. This comparison clearly illustrates the difficulty to obtain a model that performs consistently over many cases and therefore the need for further work in the definition of performant scoring functions. We've added one sentence in the discussion to highlight that fact:

“The comparison of the different methods clearly illustrates the difficulty in obtaining a model that performs consistently across the diversity of PPIs and calls for more research to engineer better featurization, datasets and scoring functions.”

Q4. As the platform is broader than ranking (PPI models) and is generic in extracting 3D structure information from protein-protein interfaces, the current name of DeepRank may be limiting a broader impact. The platform is more about data mining from 3D PPI than that for PPI model ranking.

A4. Thank you for your suggestion. We agree that the name might not be ideal and we considered changing it before submitting the paper. However, the tool is already used by different research groups, as illustrated by the number of “stars” and forks of the GitHub repository. To avoid issues for the people already using the tool we have decided to keep the name as it is.

Q5. Although GNNs and equivariant NNs were recognized in the introduction or discussion, their latest developments directly for application 2 (scoring PPI models) were not. Examples include GNN for scoring PPIs: <https://doi.org/10.1002/prot.25888> Equivariant NNs for scoring PPIs: <https://doi.org/10.1002/prot.26033>

A5. Thank you for pointing out these recent publications. We have now included the GNN reference to the text and added one sentence to illustrate the use of equivariant network:

“Finally, rotation-equivariant neural networks have recently been used by Eisman et al on point-based representation of the protein atomic structure to classify PPIs”

Q6. Please consider reporting memory comparison in the supp. 3-E in addition to time comparison.

A6. Thank you for pointing this out. We re-benchmarked DeepRank and MASIF on a new machine with a different type of CPUs. The memory comparison was added and time comparison was updated in the Table S3. It shows that MASIF requires about 48 times more time and 7 times more memory than DeepRank.

Q7. The DeepRank data processing time in Table S2 was on average 5 seconds only. As multiple sequence alignment (MSA) costs much time and is needed for PSSM features, I wonder whether such processing time did not include that for MSA. Please clarify. If MSA is additionally part of the data processing in DeepRank, please consider including more details (library, software, parameters etc).

A7. Thank you for pointing this out. Indeed, the time for the generation of MSA is not included. We have now clarified this point in section 3-E of the Supplementary Material:

“Note that MASIF uses two geometric and three chemical features. DeepRank uses 36 physico-chemical features, out of which PSSMs (accounting for 20 features) were precalculated and provided as input”.

REVIEWER COMMENTS

Reviewer #1 (Remarks to the Author):

I am happy with the revision. The authors have added some extra text to explain the redundancy reduction better, they have added more text to compare other systems better and they have introduced MCC to gauge models rather than just relying on weighted accuracy because it is an unbalanced class problem. They also agreed to move unnecessary technical details to supplement. They rephrased some paragraphs.

Reviewer #3 (Remarks to the Author):

The revision has addressed the comments properly and is deemed satisfactory.